# Habitat Suitability and Determinants for Anatidae in Multi-Watershed Composite Wetlands in Anhui, China

**DOI:** 10.3390/ani14071010

**Published:** 2024-03-26

**Authors:** Jiye Shi, Lei Meng, Shanshan Xia, Song Liu, Lizhi Zhou

**Affiliations:** 1School of Resources and Environmental Engineering, Anhui University, Hefei 230601, China; x21301111@stu.ahu.edu.cn (J.S.); x20301102@stu.ahu.edu.cn (L.M.); xss0515@stu.ahu.edu.cn (S.X.); 2Anhui Province Key Laboratory of Wetland Ecosystem Protection and Restoration, Anhui University, Hefei 230601, China; 3Anhui Shengjin Lake Wetland Ecology National Long-Term Scientific Research Base, Dongzhi 247230, China; 4Anhui General Station of Wildlife Monitoring of Epidemic Sources and Disease, Hefei 230088, China; liusong2920@163.com

**Keywords:** habitat suitability, MaxEnt, Anatidae, wildlife, epidemic sources, Anhui Province

## Abstract

**Simple Summary:**

Anatidae are not only indicators of wetland quality but also significant subjects for monitoring epidemic diseases. Revealing potential distribution sites and influencing factors is crucial in Anatidae conservation and biosecurity maintenance. Our research, conducted in Anhui Province, China, from 2021 to 2022, mapped out the preferred habitats of twenty-one Anatidae species and analysed environmental factors that affect their distribution. By utilizing the MaxEnt model, we discovered that high-suitability habitats were predominantly located in large lakes of the Yangtze River floodplain. Our findings were validated by generalized linear mixed models (GLMM), which confirmed the influence of various factors on Anatidae richness. Specifically, higher bird richness was linked to flatter terrains near farmlands, while elements such as increased elevation and human activities were linked to lower richness. Our research underlines the importance of the Yangtze River floodplain lakes for Anatidae conservation and provides valuable insights for developing effective strategies to protect these waterbirds and their ecosystems, which is crucial to biosecurity and preventing the spread of waterbird epidemics.

**Abstract:**

Habitat suitability analysis is essential in habitat and species conservation. Anatidae are known for their migratory behaviour, high population density, and wide distribution range. Understanding their habitat utilzation and influencing factors is crucial in targeted conservation and management. In this study, we collected Anatidae diversity data, including the number of species, through field surveys from October 2021 to March 2022 and thirty habitat variables through an online database in Anhui Province, China. By using MaxEnt, we simulated the habitat suitability of twenty-one Anatidae species, revealing potential distribution sites in Anhui Province. Generalized linear mixed models (GLMM) were employed to identify factors affecting the distribution of geese and ducks. The results showed that high-suitability habitats were predominantly located in the large lakes of the Yangtze River floodplain. The GLMM analysis showed significant correlations between Anatidae richness and altitude, distribution of farmland, and human footprint. In addition, ducks were more sensitive to the human interference factor than geese. In summary, the lakes in the Yangtze River floodplain emerged as the most important Anatidae habitats in Anhui Province due to their abundant wetland resources, flat terrain, and high distribution of farmlands. These findings provide a scientific basis for the development of relevant conservation strategies and measures, aiding in wildlife epidemic monitoring, prevention, and control.

## 1. Introduction

Habitat refers to the natural or physical environment in which a particular species or organism lives and depends on for various resources, such as food, shelter, water, and reproduction. Habitat loss and fragmentation have implications for biodiversity reduction. Species conservation requires not only the maintenance of population size but also the protection of habitats [1]. Scientifically based habitat assessment and conservation planning can help maximise species conservation effectiveness [2]. Understanding species–environment interactions is key to assessing the distribution, migration, population dynamics, and ecological adaptability of species [3]. Such understanding provides a theoretical basis for biodiversity conservation and habitat management.

Anatidae, which display specific adaptive features and behaviours, heavily rely on wetlands. Their diversity, widespread presence, and seasonal migratory habits make them important wetland ecological indicator species. Anatidae community dynamics closely correlate with wetland quality. Recent severe outbreaks of highly pathogenic avian influenza (HPAIV) from 2021 to 2023 have caused extensive wild bird and poultry mortality worldwide [4]. These public health crisis events highlight the need for epidemic disease source surveillance and research. Migratory birds, especially Anatidae, majorly contribute to infectious disease transmission, such as in highly pathogenic avian influenza [5]. Analysing habitat suitability provides insights into their distribution characteristics, serving as a valuable reference for Anatidae conservation and environmental management [2]. Additionally, it also provides basic distribution information for preventing epidemic infectious disease risks associated with host animals.

The MaxEnt model is one of the most popular tools for modelling species distributions and ecological niches. In comparison with traditional ecological modelling methods, the MaxEnt model has the advantages of not requiring complete sample data, being suitable for small sample sizes, and possessing high prediction accuracy [6]. Importantly, the relevance of the chosen environmental factors affects the accuracy of the MaxEnt model [7]. Many studies have predicted the distribution of Anatidae, as well as changes in their habitat location and utilisation, and in these studies, a number of ecological factors have been found to be strongly associated with the distribution of Anatidae. For example, factors such as proximity to water sources, land use type, and elevation affect the distribution of wintering Anatidae [8,9,10]. Shoals play a crucial role in supporting the survival of Anatidae through the provision of suitable food, hydrological conditions, and diverse biodiversity [11]. Human disturbances significantly impact species distribution, exerting increased pressure on these populations [12]. Agricultural fields can provide important food resources for wintering Anatidae and are also sites for migratory birds to replenish their energy [13]. Anatidae distribution prediction using the environmental factors mentioned above, as well as exploring the significance factors affecting the distribution of Anatidae, is necessary.

Anhui Province in China is home to various wetland environments, like lakes, rivers, marshes, and other wetland waters, which provide ideal habitats rich in food resources for Anatidae. These characteristics have established Anhui Province as a critical flyway habitat for many Anatidae species [14]. Anhui Province is also an essential distribution area for endangered Anatidae, such as Baer’s Pochard (*Aythya baeri*), Scaly-sided Merganser (*Mergus squamatus*), and Lesser White-fronted Goose (*Anser erythropus*). However, in recent years, Anhui Province has faced serious biosecurity threats, including avian influenza (AI), African swine fever (ASF), and coronavirus disease 2019 (COVID-19), posing serious threats to both human society and wildlife populations. Some studies about Anatidae have also been conducted in Anhui Province. An analysis of the distribution of wintering Tundra Swans (*Cygnus columbianus*) in the middle and lower Yangtze River floodplains revealed that Anhui Province is a densely populated distribution area for wintering Tundra Swans [11]. Altered foraging behaviour and habitat use by wintering White-fronted Geese (*Anser albifrons*) have been found in Shengjin Lake [15]. Habitat factors have been found to affect the foraging strategies of wintering Tundra Swans in Huangpi and Shengjin Lakes [16]. The environmental factors of different wetland habitats with wintering ducks were studied at Shengjin Lake [17]. H6 AIVs are carried by geese and have the potential to infect mammals [18]. Overall, previous research has established Anhui Province as an important distribution area for Anatidae. However, all these studies have assessed a single Anatidae species, and regional analyses of suitable habitats for all Anatidae have not yet been carried out. Moreover, the wetlands within the Yangtze, Xin’an, and Huaihe River basins of Anhui Province exhibit notable differences, with the spatial distribution of Anatidae being significantly influenced by habitat preferences. Specific wetland conditions are often favoured by these species. Consequently, comprehensive habitat suitability analyses for all Anatidae species within Anhui Province are imperative to discern their preferred habitats.

This study aimed to assess Anatidae habitat suitability in Anhui Province, predict their potential distribution, and gain insights into their habitat selection preferences in the region. The anticipated outcomes seek to offer a scientific foundation for Anatidae protection and management. Furthermore, this study aimed to broaden the scope of Anatidae research application, offer basic information on the spatial distribution of host animals for the early warning of epidemics, and contribute to the protection of biodiversity and maintenance of public health and safety [19].

## 2. Materials and Methods

### 2.1. Study Area

Anhui Province has a total area of 140,100 km^2^. The study area included various wetlands in the Yangtze, Xin’an, and Huaihe River basins in the province. These three rivers form multi-floodplain wetlands in this region. Most of the significant lakes and wetlands in Anhui are situated in the floodplains along the Yangtze and Huaihe Rivers, and the wetlands located in this area serve as crucial wintering ground for Anatidae. Additionally, many small reservoirs and ponds found between the Yangtze and Huaihe Rivers play a significant role as waterfowl habitats. The Xin’an River basin, located in the mountains of southern Anhui, primarily consists of riverine wetlands and valley-type reservoirs. The multiple river basins and unique landscape have created rich wetland resources and diversified wetland types in Anhui Province. Annually, over 200,000 waterbirds along the East Asian-Australasian Flyway choose to winter here [20].

Anhui Province experiences a subtropical monsoon climate with an altitude range of −3–1864 m. The average annual temperature ranges between 14 °C and 17 °C, with an average winter temperature of about 4 °C. The average annual precipitation range is 773–1670 mm, mainly concentrated in the spring and summer [21].

### 2.2. Occurrence Data of Anatidae

Anatidae occurrence data in Anhui Province were gathered through extensive field surveys, during the wintering period from October 2021 to March 2022. Number and distribution of Anatidae in Anhui were obtained from a synchronized survey in Anhui Province during the overwintering period in January 2022. A wetland unit with a number > 30,000 Anatidae or a distribution > 15 species of Anatidae can be considered a hotspot for the distribution of these species. To comprehensively assess Anatidae presence in each wetland, this study adopted a fixed-point counting method. For smaller wetlands offering a wide field of view, observation sites were strategically chosen with fewer obstructions, or higher terrain was selected as the observation point. Conversely, large wetlands or those with limited visibility had the number of survey points selected according to their size, concentrations of waterbirds, and a wider field of view. To avoid double-counting, distances exceeding 2 km were chosen between neighbouring observation points. Identifying waterbirds to the species level and counting them within the effective monocular field of view (1 km) (ATS 20–60 × 80; Swarovski, Absam, Austria), took approximately 20 min per monitoring site, ensuring precision in data collection. The counting method varied based on waterbird population density: direct counting for areas with small numbers and group counting for large numbers.

The substitution of Anatidae occurrence points into the MaxEnt model requires preprocessing. The distribution records of each Anatidae species were meticulously screened, eliminating offset values and redundant data, ensuring model accuracy and eliminating model bias caused by spatial autocorrelation [22]. Only Anatidae species with over 10 distribution records were selected as research objects, and in each grid of approximately 1 km^2^, only one distribution point record was maintained for each species, this preprocessing operation was performed in R 4.2.3 using the “ENMTools 1.1.2” package [23,24]. Finally, the screening process identified 21 Anatidae species across 2169 distribution points (Figure 1).

### 2.3. Environmental Variables

Based on previous studies, data on 26 environmental variables potentially relevant to the distribution of Anatidae were collected (Appendix A). This dataset included 19 bioclimatic variables which are known for their ability to simulate recent climatic conditions in the region and are extensively utilized in species distribution modelling [8] (http://www.worldclim.org, accessed on 22 May 2023). Land use classification data were sourced from the Resource and Environment Science and Data Center, employing Landsat remote sensing imagery from 2023 as the primary information source [9]. Through remote sensing interpretation, a national-scale land use thematic database was constructed (https://www.resdc.cn, accessed on 23 May 2023). Population spatial distribution data were acquired from the Resource and Environment Science and Data Center (https://www.resdc.cn, accessed on 24 May 2023). These data, based on national sub-county population statistics, incorporate multiple factors, such as land use type, nighttime light intensity, and settlement density, all of which are significantly associated with population dynamics. A multi-factor weight allocation method was applied to distribute the population data across a spatial grid, with administrative districts serving as the basic statistical units, thus enabling the spatialization of the population distribution [10]. Digital Elevation Model (DEM) data were procured from the Geospatial Data Cloud (http://www.gscloud.cn, accessed on 25 May 2023). For the study area, data with similar dates and less than 5 percent cloud cover were selected and adjusted. Data on main roads were acquired from OpenStreetMap (https://www.openstreetmap.org, accessed on 27 May 2023). After having received the vector data for roads and settlements, the Spatial Analysis Tool in ArcGIS 10.7 was utilized to calculate the raster data representing distances from these features within the study area [11]. Farmland distribution data were procured from the dataset on China’s farmland distribution published [25] (https://doi.org/10.5281/zenodo.7936885, accessed on 21 December 2023), which was refined through a classification procedure reliant upon remote sensing images for the corresponding year, enhanced by machine learning algorithms. Human footprint data were derived from a dataset developed [26] (https://www.x-mol.com/groups/li_xuecao/news/48145, accessed on 23 October 2023). This dataset incorporates eight variables indicating varied dimensions of human impact, such as the built environment, population density, pastureland, roads, railways, and navigable waterways. The methodology adheres to that established by Sanderson and Venter for analysing annual dynamics of the global terrestrial human footprint.

### 2.4. Construction and Evaluation of MaxEnt

Many environmental variables are spatially autocorrelated, and multicollinearity between variables can lead to the overfitting of species distribution models [27]. To improve the accuracy of the model prediction results, all environmental variables must be screened and tested for relevance. Initially, all environmental variables were simulated in the MaxEnt model, version 3.4.4 (https://biodiversityinformatics.amnh.org/open_source/maxent/, accessed on 29 May 2023), to determine their contribution [28]. Subsequently, the “ENMTools” package in R, version 4.2.3, was used to calculate the Spearman correlation coefficients between variables. If the absolute coefficient value exceeded 0.8, only variables with a greater initial contribution were retained [29]. The specific variables retained post-autocorrelation analysis for each species are listed in Appendix A.

The latitude and longitude information for each of the 21 species and filtered environmental variables was loaded into MaxEnt for modelling. The model randomly allocated 75% of the point data for training, using the remaining 25% as a test set and employing the knife-cut method for validation. We determined the appropriate number of iterations based on the convergence of the model or through cross-validation. This process was repeated 10 times using the “Bootstrap” run type, with a maximum number of iterations set to 500 per run. The final prediction result of the species distribution was derived from the average output of the 10 sub-models per pixel [30]. Model accuracy was assessed by employing receiver operating characteristic (ROC) curves to determine the accuracy of the model [31]. An area under the ROC curve (AUC) value between 0 and 0.6 indicates invalid model predictions; a value within 0.6–0.7 implies poor accuracy; a value within 0.7–0.9 suggests moderate accuracy; and a value > 0.9 indicates high accuracy, with values approaching 1 reflecting strong predictive power [32]. However, it is important to note that in environmental data modelling, an AUC value of exactly 1 may be exceptionally rare and could suggest the artificial fitting or overfitting of the model to the training data. Therefore, while high AUC values close to 1 are desirable for demonstrating good model performance, exceedingly high values should be scrutinized for potential overfitting. Anatidae richness > 15 is considered a highly suitable habitat; between 6 and 15, it is a moderately suitable habitat; a value of 1 to 6 suggests a lowly suitable habitat; and a value of less than 1 indicates an unsuitable habitat [8].

Because each species exhibits a different level of environmental tolerance, suitable habitat thresholds were determined based on the maximum available distribution records for each species. The habitat suitability for each sampling site was extracted from habitat suitability maps calculated using the model. The standard deviation (σ) and the mean value (μ) were calculated according to the theory of normal distribution; μ − σ was selected to transform species probability distribution maps into 0/1 binary distribution maps. Finally, these binary distribution maps for 21 species were superimposed and merged using ArcGIS to produce the predicted potential distribution maps of Anatidae [33].

### 2.5. Statistical Analysis

To explore the factors influencing habitat suitability for Anatidae, we extracted the 26 environmental variables at each survey location. Anatidae were divided into geese and ducks for analysis due to their different food and habitat needs. These data were then exported to the loci for GLMM analysis [34] in R 4.2.3.

Before regression analysis, we standardised all explanatory variables and calculated variance inflation factor (VIF) values for each variable. We progressively removed environmental variables with the highest VIF values until the condition of VIF < 2 was met [35]. Ultimately, certain factors—distribution of the population, distance from the countryside, human footprint, altitude, distance from roads, and distribution of farmland—were used as fixed effects. Wetland units were considered random effects with a Poisson distribution, and only the main effects of the model were considered [36]. To construct a multivariate generalised linear mixed-effects model, all predictor variables were considered potential fixed factors. By using the “MuMIn 1.47.5” package in R, a full set of models was generated using the dredge function. The model was used to generate a set of predictor variables for each variable. These models were ranked using the Akaike Information Criterion (AICc) corrected for small sample sizes, and the model with the lowest AICc value was selected as the best model [37]. Additionally, to address multicollinearity between different explanatory variables, the data matrix was randomised 10,000 times. The significance of each predictor was determined by summing the Akaike weights (ΣWi) in the set of models where the variable was present. We considered a variable to be consequential if the importance value was >0.5, meaning that half or more of the total Akaike weight for the model set was represented by models that contained that variable [38]. We estimated the independent and joint contributions of each variable and identified statistically significant variables using the “glmm.hp 0.1–2” software package [39]. The above operations were performed using R 4.2.3.

## 3. Results

### 3.1. Number and Distribution of Anatidae in Anhui

In Anhui Province, a total of 30 Anatidae species were recorded in the three major river basins, totalling 537,902 birds. The population distribution of Anatidae in the Yangtze, Huaihe, and Xin’an River floodplains accounted for 76.32%, 23.46%, and 0.02% of all Anatidae, respectively (Figure 2). The top ten Anatidae species, in descending order by population size, included Bean Goose (*Anser fabalis*), Falcated Duck (*Mareca falcata*), Eastern Spot-billed Duck (*Anas zonorhyncha*), Green-winged Teal (*Anas crecca*), Mallard (*Anas platyrhynchos*), Graylag Goose (*Anser anser*), Tundra Swan, Baikal Teal (*Sibirionetta formosa*), Common Pochard (*Aythya ferina*), and White-fronted Goose (*Anser albifrons*). These species collectively accounted for 96.06% of the total Anatidae count (Figure 3). Key Anatidae hot spots included Shengjin, Caizi, Wuchang, and Huang Lake in the Yangtze River floodplain and Chengdong, Qili, and Chengxi Lake in the Huaihe River floodplain.

### 3.2. Habitat Suitability for Anatidae in Anhui Province

The mean AUC value of the distribution model ROC curve for the 21 species was 0.944 with a standard deviation of 0.024 (Appendix A), indicating that all MaxEnt-simulated models accurately predicted the distribution ranges of Anatidae, indicating a strong fit and plausible results (Appendix A).

The predicted distribution results based on species distribution points delineate the Anatidae habitat in Anhui Province into three categories: highly suitable, moderately suitable, and lowly suitable, covering approximately 836 km^2^, 4187 km^2^, and 21,855 km^2^, respectively. Anatidae populations are primarily distributed in the Huaihe, Yangtze, and Xin’an River floodplains. Highly suitable habitats are mainly located in the Yangtze River floodplain lakes, with over 15 Anatidae species. Moderately suitable habitats are concentrated in the lakes of the Huaihe River floodplain, the southern Anhui Mountains area, and the hilly area between the Yangtze and Huaihe Rivers, accommodating 6–14 Anatidae species. Lowly suitable habitats are located in certain riverine wetlands in the Xin’an River floodplain, hosting 1–5 Anatidae species (Figure 4).

### 3.3. Factors Influencing Habitat Suitability for Anatidae

The GLMMs for geese and ducks each generated 64 candidate models. All explanatory variables included in the model analysis with the sum of Akaike weights > 0.5. All candidate models were ranked using ΔAICc (Difference between Candidate and Most Parsimonious Model), and the candidate models with ΔAICc < 2 are listed in Table 1. Based on the most parsimonious model, the GLMM showed a significant correlation between geese species richness and human footprint (HF), DEM, and distribution of farmland (DF). Meanwhile, duck species richness had significant correlations with distribution of the population (POP), DEM, distance from the countryside (CD), human footprint (HF), and distribution of farmland (DF) (Figure 5).

In the analysis of the GLMM for geese, DEM and distribution of farmland (DF) had high relative contributions of 52.582% and 31.985%, respectively, and human footprint (HF) had a low relative contribution of 15.433%. In the analysis of the GLMM for duck, DEM, distribution of farmland (DF), distance from the countryside (CD), and distribution of the population (POP) had high relative contributions of 45.629, 12.671, 15.311, and 15.222%, respectively, and distance from roads (RD) and human footprint (HF) had low relative contributions of 2.081 and 9.086% (Table 2).

## 4. Discussion

### 4.1. Model Performance

Spatial heterogeneity is strongly correlated with species distribution patterns and habitat suitability [40]. The increase in the sample size and expansion of the sampling area allowed the model to capture the distribution patterns of species more comprehensively under different environmental conditions, resulting in more reliable predictions. By conducting extensive field surveys across varied habitats in Anhui Province and collecting data at Anatidae occurrence sites, this study aimed to mitigate the influence of spatial heterogeneity on predictions, thus improving the accuracy of the species distribution model [41]. The average AUC value of 0.944 calculated from the ROC curve of the distribution model indicated that all MaxEnt-simulated models predicting the range of Anatidae were well fitted, and the results were credible. The AUC value is a commonly used indicator for assessing the performance of MaxEnt models, and a value closer to 1 indicates better predictive ability of the model [42]. Therefore, our results indicate that the MaxEnt model performed effectively in predicting the Anatidae distribution.

### 4.2. Habitat Suitability and Influential Factors of Anatidae

Understanding the relationship between species and their environment is crucial to studying their ecological requirements and spatial distribution. In this study, Anatidae richness varied with environmental factors [43]. Digital elevation, human footprint and distribution of farmland are important factors affecting the distribution of Anatidae species, and some of the factors do not affect geese and ducks to the same degree.

Anatidae usually inhabit diverse water bodies, like lakes, rivers, marshes, and others, thriving on a diet that includes aquatic plants, algae, insects, fish, crustaceans, and other aquatic animals. Shoals are essential habitats for benthic organisms, molluscs, and plankton and supply a rich and diverse range of food resources [44]. Shoals create a safer and more comfortable overwintering environment for Anatidae [45]. In addition, some studies have shown that shoals had a greater impact on the distribution of geese than ducks, mainly due to the fact that geese are more dependent on shoals, which they require to a large extent for their avoiding predation risks and nutritional supply [46]. The abundance and availability of food resources are critical factors influencing the distribution of Anatidae. Rice, the main food crop in the middle–lower Yangtze River floodplain, is harvested in autumn, and its second crop occurs after harvesting. Residual grains and seedlings are the alternative food for geese, and winter farmland provides higher foraging rates than natural mudflats and grasslands [15]. In addition, farmland of paddy and wheat fields can provide waterbirds with substitutes for natural wetlands and shelter in years of food shortage [13]. This also coincides with the results of the study, where the effect of farmland distribution on Anatidae richness was significantly and positively correlated. In summary, geese are able to search for food resources in a variety of habitats, particularly specific habitats such as farmland and shoals, and have a wider range of adaptations in foraging habitats relative to ducks.

Plant coverage is an important indicator for predicting herbivore habitat distribution [46] and is also significantly correlated with the richness of Anatidae [47]. However, this relationship is not straightforward due to the complex dietary preferences of Anatidae [48]. Dabbling and diving ducks prefer habitats with lower vegetation cover and more open water [47], whereas herbivorous and tuber-feeding geese prefer areas with relatively denser vegetation cover that are direct or indirect food cues [49]. Different types of wetland vegetation can affect Anatidae differently. In summary, feeding and habitat use contributed to the complexity of the effects of the vegetation factors on the distribution of Anatidae.

DEM plays an important role in bird habitat prediction [48]. Flat terrains allow shorter flight distances and lower energy expenditure in the search for food [47]. In the present study, areas with high Anatidae richness were concentrated between 0 and 200 m in altitude. As altitude increased, Anatidae species richness decreased. This trend aligns with the concentration of middle and high Anatidae richness in flat wetlands, like those in the Yangtze River and the Huaibei floodplain, typically at elevations from 0 to 50 m. In contrast, moderate- and low-Anatidae-richness areas were located at higher elevations in the western and southern Anhui mountain areas, which have higher average elevations, with corresponding elevations ranging from 100–300 m to 400–1000 m. Lower elevations provide a more suitable habitat for Anatidae due to their flat terrain and larger catchment areas. Also, a small amount of artificial habitat exists in high-altitude mountainous areas that can provide food for Anatidae, such as farmland and lotus root ponds. In conclusion, Anatidae tend to favour low-altitude and flat areas [47].

Human footprints, indicative of anthropogenic activities such as construction, transportation, energy consumption, and waste production, are critical traces that facilitate the understanding of the environmental and ecological consequences of human presence [49]. Rigorous analyses of human footprint data can elucidate the extent to which these activities disrupt ecosystems. There is accruing evidence that recreational pursuits may adversely affect waterfowl populations, just as fishing activities have been shown to negatively impact wading bird species [50]. However, these are not the sole instances of anthropogenic influences on the Anatidae family. Further studies illustrate that the implications of human activity extend beyond leisure and fishing, and including serious habitat disruption due to wetland development and land use changes, leading to fragmentation and habitat loss, which pose significant threats to the habitats of Anatidae species [24]. Additionally, the adverse effects of water pollution, river diversions, and water level controls in the wake of industrialization and urbanization cannot be overlooked [51]. Given the multifaceted nature of human impacts on the Anatidae, it is imperative to engage in a holistic appraisal of these pressures to inform effective conservation and management strategies for the protection of Anatidae populations [52]. It is clear that sustaining the biodiversity of these waterfowl necessitates an integrated approach taking into account the comprehensive spectrum of human-induced environmental changes.

### 4.3. Protection Measures and the Epidemic Disease Source Prevention of Anatidae

This research has produced a potential Anatidae distribution map, crucial to predicting their present and future locations. It aims to guide conservation efforts by providing recommendations to conservation managers. Considering that Anhui Province is an important migratory stopover and wintering site for Anatidae, there is an urgent need to formulate practical conservation policies for these migratory species [13]. Firstly, it is necessary to establish nature reserves in shallow river-connected lakes in the middle and lower Yangtze River floodplain, as well as in Xiangjian and Huangpi Lake. Addressing conservation gaps and creating buffer zones between urban and moderate habitat suitability zones are critical steps. For example, small reservoirs and ponds generally have high levels of human activity that are not conducive to Anatidae habitat. These actions are intended to limit human activities in these areas to ensure the safety and protection of Anatidae [11]. To effectively promote conservation, it is necessary to increase investment and manpower in highly suitable habitat areas. Simultaneously, establishing an Anatidae ecological monitoring network is essential. Regularly monitoring Anatidae numbers, distribution, and migration will enable prompt issue detection and the implementation of necessary conservation measures [53].

The effective management of epidemic disease source surveillance is essential in biodiversity conservation and social development. Strengthening the monitoring of highly suitable habitat areas during Anatidae migration and the wintering period is crucial. Immediate actions should follow any abnormal deaths to achieve early detection, timely prediction, and an effective response to control epidemics. When recognising the spatial correlation between migratory bird routes and epidemic outbreaks in Anhui Province, it is important to identify and monitor migration routes and ecological corridors to ensure biosecurity [5]. Given that poultry farming significantly impact avian influenza occurrence, avoiding poultry and wild bird mixflocks in moderate- and high-suitability habitat zones is critical. The Anatidae aggregation should prohibit poultry farming to prevent epidemic spread to humans, poultry, and livestock [54].

## 5. Conclusions

This study systematically investigated and analysed habitat suitability for Anatidae in Anhui Province and the factors influencing it. By using the MaxEnt model and GLMM analysis, we mapped the potential Anatidae distribution in Anhui Province. Our findings highlight elevation, human footprint, and distribution of farmland as major factors affecting habitat suitability. In addition, ducks were more sensitive to the human interference factor than geese. The lakes in the Yangtze River floodplain, Shengjin, Caizi, and Chenyao Lakes, emerged as the most suitable habitats for Anatidae during winter. Lakes in the Huaihe River floodplain exhibited medium suitability as wintering destinations for these species. These results lay the foundation for the habitat restoration and ecological management of Anatidae wintering sites. Prospective potential distribution maps can also be used to target medium and highly suitable habitats for the prevention and control of epidemic disease sources. These findings will help better identify Anatidae aggregation areas and improve epidemic disease source control for humans and waterbirds.

This study used 2021–2022 data, suggesting potential improvements with an additional field survey for model enhancement. While we have focused on environmental factors like digital elevation, human footprint, and distribution of farmland, other aspects, like water quality, level, and food resources, remain unexplored, and the complex effects of different vegetation types on Anatidae have not yet been elucidated, needing further investigations [45,50,55]. Future studies should address how Anatidae habitats intersect with human activity due to climate change and human development. Therefore, future research should focus on the spatial and temporal distribution of Anatidae, their habitats, and infections to protect Anatidae populations and maintain public health safety.

## Figures and Tables

**Figure 1 animals-14-01010-f001:**
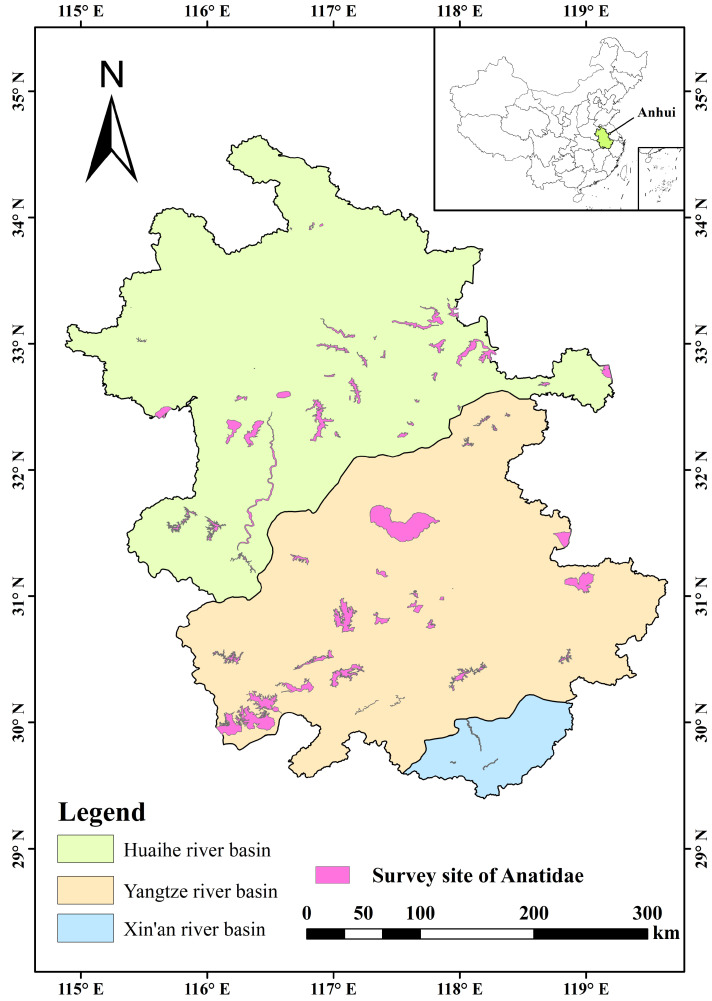
Study area with survey site of Anatidae in Anhui Province.

**Figure 2 animals-14-01010-f002:**
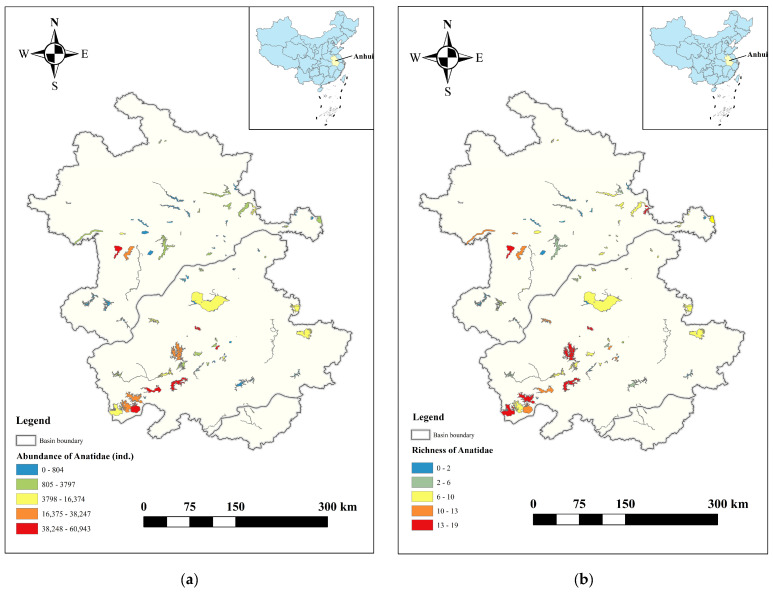
Distribution of abundance and richness of Anatidae in Anhui Province. (**a**) Abundance, (**b**) Richness.

**Figure 3 animals-14-01010-f003:**
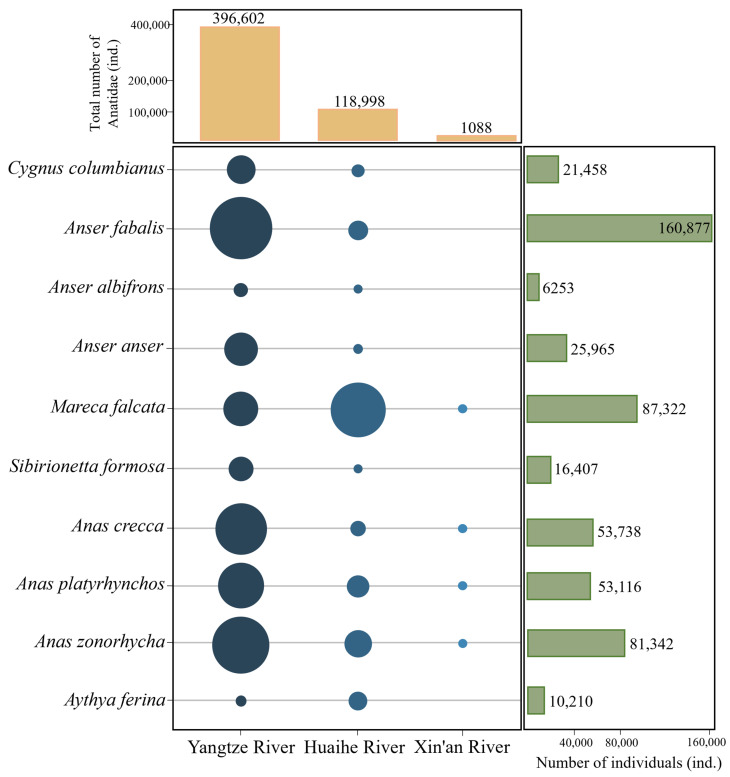
Number and distribution of top ten Anatidae species in the Yangtze, Huaihe, and Xin’an River floodplains.

**Figure 4 animals-14-01010-f004:**
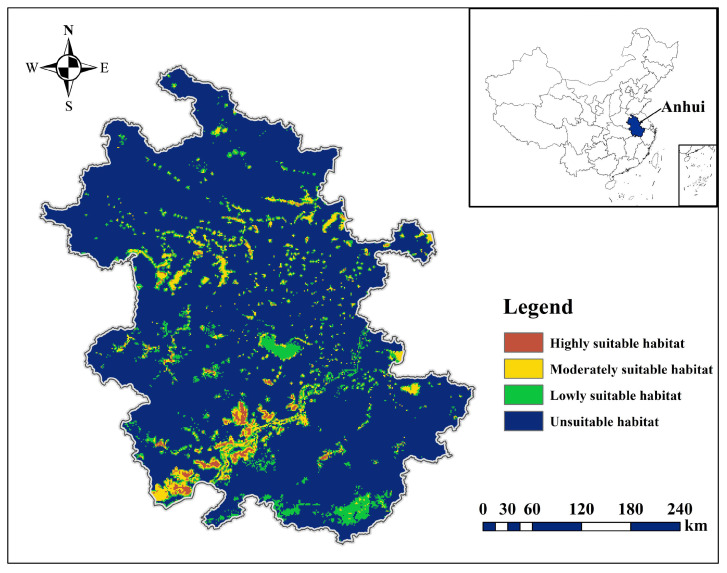
Potential distribution of Anatidae in Anhui Province.

**Figure 5 animals-14-01010-f005:**
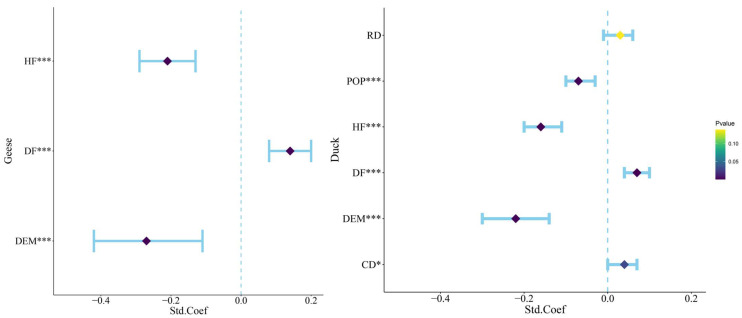
Significant factors influencing the distribution of Anatidae. The blue intervals represent the range of 95% confidence intervals for each factor. *** indicates factors with 0 < *p* < 0.001 and highly significant correlation, ** indicates factors with 0.001 < *p* < 0.01 and moderately significant correlation, and * indicates factors with 0.01 < *p* < 0.05 and significant correlation. DEM: digital elevation model; HF: human footprint; DF: distribution of farmland; POP: population data; CD: distance from the countryside; RD: distance from roads.

**Table 1 animals-14-01010-t001:** Explanatory variables for ΔAICc < 2 and Anatidae species richness models.

Response Variable	Explanatory Variables	df	logLik	AICc	ΔAICc	Weight
Species richness of geese	DEM + DF + HF	6	−1252.8	2515.723	0.00	0.424
	DEM + DF + HF + CD	7	−1252.6	2517.319	1.65	0.221
	DEM + DF + HF + RD	7	−1252.7	2517.626	1.90	0.232
	DEM + DF + HF + POP	7	−1252.7	2517.634	1.91	0.216
Species richness of duck	DEM + DF + HF + CD + POP + RD	8	−2353.2	4722.651	0.00	0.436
	DEM + DF + HF + CD + POP	7	−2354.3	4722.792	1.65	0.415
	DEM + DF + HF + POP + RD	7	−2355.5	4722.912	1.90	0.176

Models with ∆AICc > 2 are not shown. DEM: digital elevation model; HF: human footprint; DF: distribution of farmland; POP: population data; CD: distance from the countryside; RD: distance from roads. df: degrees of freedom; logLik: value of the log-likelihood function; AICc: calibration value of the Akaike Information Criterion (AICc), ΔAICc: difference between the AICc values; weight: weight of the model.

**Table 2 animals-14-01010-t002:** Degree of contribution to population variance by explanatory variables in the GLMMs.

Response Variable	Explanatory Variables	Unique	Average Share	Individual	I.perc (%)
Species richness of geese	DEM	0.209	0.028	0.237	52.582
	DF	0.010	0.010	0.020	31.985
	HF	0.009	0.014	0.023	15.433
Species richness of duck	DEM	0.280	0.025	0.304	45.629
	DF	0.005	0.012	0.017	12.671
	HF	0.020	0.013	0.034	9.086
	CD	0.004	0.004	0.008	15.311
	POP	0.003	0.003	0.001	15.222
	RD	0.001	0.007	0.008	2.081

DEM: digital elevation model; HF: human footprint; DF: distribution of farmland; POP: population data; CD: distance from the countryside; RD: distance from roads.

## Data Availability

The data supporting the results of this article will be available by the authors on request.

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
