# Peer review of "Habitat Suitability and Determinants for Anatidae in Multi-Watershed Composite Wetlands in Anhui, China"

_animals, 2024, doi:10.3390/ani14071010_

Round 1

Reviewer 1 Report

Comments and Suggestions for Authors

The authors used MaxEnt and GLMM to explore the potential distribution and driving factors of Anatidae in Anhui Province based on filed survey data of waterbirds in 2021-2022 and environmental variables. Their results indicated that high suitable habitats were distributed in Yangtze River. They also identified the major factors influencing the abundance of Anatidae. The study provides a scientific basis for the development of conservation measures. This is an interesting paper with scientific significant. However, there are several shortcomings.

 Major comments:

The food compositions and habitat requirement of geese and ducks are different. Geese mainly feed fresh grass leaves on meadows, while ducks mainly feed aquatic plant seed and invertebrate in waters. I suggest you divide Anatidae to ducks and geese, or seed eaters, sedge eaters and tuber eaters, or other kinds of groups according to their diet and habitat requirement, and reconducted the GLMM analyses. Because the influencing factors of the abundance of ducks and geese should be different or even contrary. Integrating the two waterbird groups would also make your results difficult to explain.

You only included the potential distribution area predicted by MaxEnt. I suggest you to add the ACTUAL distribution map of Anatidae in the text. Because waterbird distribution hop spot and conservation priority should be identified based on the actual distribution instead of predicted distribution.

 Your protection recommendations in the Discussion are too broad, and should be more specific according to your results. For example, you recommend “establishing nature reserves in high habitat suitability zones, addressing conservation gaps, and creating buffer zones between urban and moderate habitat suitability zones are critical steps”, but how? According to your results, where should we build nature reserve, where is the conservation gaps? You also recommend “it is necessary to increase investment and manpower in highly suitable habitat areas”, but where is the highly suitable habitat areas? Be clearer. We can give these recommendations without your study; thus, you should give specific suggestion based on your results.

 Other small comments:

Line 26: This sentence lacks a verb.

Line 29: You mentioned in the Method that the environment variables came from online database instead of collected by your field survey. Please correct it.

Line 28-29: You should clarify your survey months, so that readers can know your target waterbird groups, i.e., wintering waterbirds, breeding waterbirds, or migrates.

Line 79: Change the period between “distribution” and “excerting” to comma.

Lune 132: October to March is the wintering period of waterbirds, not the migration period. Thus, your study mainly focused on wintering waterbirds. You’d better change it through the main text.

Lines 130-143: How often and how many times did you conduct field survey? Just one time or several time? Was the survey synchronous or nonsynchronous? These information are critical, and you should clarify them in the Method.

Figure 1: Mark the positions of the three floodplains (i.e., the Yangtze, Xin’an and Huaihe river floodplalin) in this figure.

Figure 2: Do you mean factors USED in the GLMM? Add a verb.

Line 223: How did you get the total number data? By adding abundance of each species within a period or the whole wintering period?

Line 228: Delete the Latin name of Tundra Swan as you only needed to show the Latin name the first you mentioned a new species.

Line 231-233: How did you define hot spots? By total bird number access a threshold or other method? Clarify it.

Figure 3: Add the “top ten Anatidae species” in the legend.

Line 239: Except for mean value, you also need to show the standard deviation.

Figure 4: I suggest you to replace the ROC curves with the potential distribution map of each bird species predicted by Maxent, which is more important information. You could show the mean vales (± SD) of ROC in the main text and move the ROC figure to appendix.

Lines 245-246: How did you define the three categories of suitability? What’s the thresholds? You seem not clarify it in the Method.

Line 258: In Table 1, you listed models with delta AIC < 3 instead of < 2. I wonder what the threshold did you really used? 2 or 3?

Line 260: DEM and NDVI are commonly used short names, thus, readers can easily understand. However, HF, SD and DF are defined by yourself, which is not friendly to reads. I suggest you to use full names of HF, SD and DF since the full names are not long, and most important, easily for readers to understand.

Lines 260-261: I suggest you to add “because its 95% confidence intervals contained 0” after “POP were not statistically significant”.

Lines 269-271: Since you used so many short names, I need to check its meaning in the previous text every time. That’s inconvenience. Therefore, full names should be better.

Table 2: Unify decimal point in this table.

Lines 275-278: What does this paragraph mean? It is not related to your study.

Line 303: Your study objects are wintering waterbirds, and the requirement of waterbird breeding environment is not related to your study.

Reviewer 2 Report

Comments and Suggestions for Authors

The manuscript provides interesting information about the distribution models of waterfowl in a region of China. However, it presents some issues that need to be addressed.

Materials and Methods

The authors need to provide more detailed information about the methodology used.

Line 131-132. Regarding the conducted counts, they indicate that the census period extended between October 2021 and March 2022. During that period, the habitat preferences of species and their abundance can change considerably. Were all wetlands in the region surveyed every month? If so, which counts were used for modeling with Maxent? If not all wetlands in the region were surveyed each month, what counts were used for modeling?.

Line 148-149. The authors indicate that to avoid autocorrelation, one point per 1 x 1 km grid was used for each species. For species with more than one point available per grid, the authors should indicate how the point used in the modeling was chosen.

Line 211-213. The authors must specify the following issues:

- What type of model they used (only main effects or also interactions)?

- How many models they generated.

- It would be necessary for them to determine the importance of each predictor by summing the Akaike weights (ΣWi) in the set of models where that variable was present. In this way, consecutive predictors would be those for which ΣWi > 0.5 (Taylor and Knight 2003).

- I recommend using other references to document this procedure, e.g. (Burnham and Anderson 2002, Burnham and Anderson 2004, Burnham et al. 2011).

Results

Line 223. The global number of birds (537,902) is understood to be the sum of those censused throughout the study period (October 2021 to March 2022). Is it like that?. It would be necessary to know the number of birds censused simultaneously (for example in a certain month) in the region. Is this possible?.

Line 256. In this section the authors must indicate how many candidate models they generated.

Line 258. The authors state that they selected the candidate models with ΔAICc < 2 but in Table 1 they mention candidate models with ΔAICc < 3. Review this statement.

Discussion

The discussion seems acceptable to me in its current version.

References

Burnham, K. P., and D. R. Anderson. 2002. Model selection and multimodel inference: a practical information-theoretic approach. Springer-Verlag, New York.

Burnham, K. P., and D. R. Anderson. 2004. Multimodel inference - understanding AIC and BIC in model selection. Sociological Methods & Research 33:261-304.

Burnham, K. P., D. R. Anderson, and K. P. Huyvaert. 2011. AIC model selection and multimodel inference in behavioral ecology: some background, observations, and comparisons. Behav Ecol Sociobiol 65:23-35.

Taylor, A. R., and R. L. Knight. 2003. Wildlife Responses to Recreation and Associated Visitor Receptions. Ecological Applications 13:951-963.

Reviewer 3 Report

Comments and Suggestions for Authors

This is an interesting paper, generally well prepared. My doubts relate to the fact based on the parameter of "presence" of birds, without taking into account their numbers can be very misleading. The concentrations of species and the main wintering grounds, and the place where there are single individuals are treated the same.  This is not verified by the very short period of data collection, as only one season. So seasonal fluctuations in water levels, water management, rainfall, crops, human activity could have had a very significant impact on the results.

The second doubt concerns the 30 environmental parameters, were they up to date for the time of the study or did they come from many years ago? Because if the description of the environment came from a different period than the duck counts z then the accuracy of the results becomes questionable.

Comments on the Quality of English Language

The quality of English language is good.

Round 2

Reviewer 1 Report

Comments and Suggestions for Authors

I did not find big problems in the revision. However, I notice that revised context is really difficult to understand, much more difficult than the original draft. I don’t understand most part of the revised context. I strongly suggest you to carefully revise the language. It’s better to find an English native speak to polish the language.

 Line 28: It is Anatidae diversity data, not distribution data.

Line 33: Change “geese and duck species” to “geese and ducks”.

Lines 105-108: This sentence is difficult to understand. I don’t get it.

Lines 205-207: What is “Anatidae abundance” mean? As I know Maxent evaluate the probability of species occurrence (0-1) instead of abundance of each grid.

Lines 218-219: I don’t get the meaning of this sentence. Please revise it.

Line 220: It should be “Anatidae was divided into geese and ducks” instead of “Geese and ducks were divided into two separate categories”.

Line 222: The GLMM analysis was conducted in ArcMap? Are you sure?

Line 246: Delete “are”.

Figure 4: It is better to move this figure to Appendix.

Lines 288-292: What is goose/duck species richness mean? I don’t understand. The independent parameter used in GLMM is species number or abundance? Be clearer.

Line: 290: What’s “relatively significant correlation” mean?

Lines 290-293: This sentence is really difficult to understand.

Lines 290, 292, and 304: Change “formland” to “farmland”.

Lines 299-301: What is the meaning of “more significant correlation”?

Line 353: What is the meaning of “geese prefer farmland habitats to ducks”?

Line 404-408: These sentences are difficult to understand.

Comments on the Quality of English Language

Very poor language, and should be improved.

Reviewer 2 Report

Comments and Suggestions for Authors

The authors have adequately incorporated most of the comments suggested in the previous version.

Minor Comments

Line 285. The authors should explain why in their cover letter they mention that “The GLMM models for geese and ducks each generated 256 candidate models” but here they only mention 32 candidate models.

Line 287. What is the optimal model? Is it the one with the lowest AIC? I suggest replacing the concept of "optimal model" with that of "most parsimonious model." Furthermore, I suggest that at least the values of the variables with the sum of Akaike weights > 0.5 be indicated.

Line 294. Table 1. The ΔAICc statistic must include two decimal places and the Akaike weight three decimal places.

Line 295. Replace “Models with ΔAICc < 3 are not shown” with “Models with ΔAICc > 3 are not shown”.
